# Synchronously Mature Intersex Japanese Flounder (*Paralichthys olivaceus*): A Rare Case

**DOI:** 10.3390/ani14202948

**Published:** 2024-10-12

**Authors:** Tian Han, Wei Cao, Lize San, Zixiong Xu, Guixing Wang, Zhongwei He, Yufeng Liu, Yuqin Ren, Yufen Wang, Xiaoyan Zhang, Jilun Hou

**Affiliations:** 1China State Key Laboratory of Mariculture Biobreeding and Sustainable Goods, Beidaihe Central Experiment Station, Chinese Academy of Fishery Sciences, Qinhuangdao 066100, China; hantian0309@163.com (T.H.); caow@bces.ac.cn (W.C.); sanlz@bces.ac.cn (L.S.); xuzx@bces.ac.cn (Z.X.); wanggx@bces.ac.cn (G.W.); hezw@bces.ac.cn (Z.H.); liuyf@bces.ac.cn (Y.L.); renyq@bces.ac.cn (Y.R.); wangyf@bces.ac.cn (Y.W.); 2Hebei Key Laboratory of the Bohai Sea Fish Germplasm Resources Conservation and Utilization, Beidaihe Central Experiment Station, Chinese Academy of Fishery Sciences, Qinhuangdao 066100, China; 3China Ocean College, Hebei Agricultural University, Qinhuangdao 066009, China

**Keywords:** synchronously sexual maturity, 21-hydroxylase activity, gene expressions, hormonal endocrinology

## Abstract

**Simple Summary:**

This study reports an unusual case of hermaphroditism in Japanese flounder captured from the Bohai Sea. In this study, the results showed that the heterozygosity of the intersex Japanese flounder was 0.632, with the synchronous maturation of the testis and ovary, and eggs and the sperm were capable of fertilization. The levels of reproduction-related hormones were intermediate; the activity of 21-hydroxylase was reduced by approximately 20.0%.

**Abstract:**

Japanese flounder is usually gonochoristic, with gonads that are either testes or ovaries. Here, we report an unusual case of hermaphroditism in Japanese flounder captured from the Bohai Sea. In the intersex flounder, the membrane of the upper ovary was closely connected to the abdominal muscles and internal organs, and the eggs filled the entire abdomen. The lower ovary was small and closely connected to the testes. The testes contained few fully mature sperm. Both eggs and sperm were capable of fertilization. The levels of several reproduction-related hormones (17β-estradiol, 11-ketotestosterone, 17α, 20β-dihydroxyprogesterone, luteinizing hormone, follicle-stimulating hormone, and testosterone) in the intersex flounder were intermediate, between those in females and males. The results showed that the heterozygosity of the intersex flounder was 0.632, and there were 28 single-nucleotide polymorphisms in the *cyp21a* gene. Compared with that of wild flounder, the activity of 21-hydroxylase was reduced by approximately 20.0%, and expressions of *cyp19a*, *amh*, and *dmrt1* differed. We present the first report of its kind, detailing the anatomy, hormonal endocrinology, molecular biology, and physiology of the intersex Japanese flounder.

## 1. Introduction

Teleosts are diverse and exhibit various vertebrate sexualities, including gonochorism and hermaphroditism [1]. Approximately 2% of teleost fish are hermaphroditic [2]. Hermaphrodism may be sequential (switching from male to female or vice versa) [3] or simultaneous (possessing both male and female characteristics) [4]. Hermaphroditic animals may have separate gonads (ovaries and testes) within their bodies, or gonads containing both tissue types in a single organ [5]. However, in gonochoristic fish, the gonads differentiate into ovaries or testes and remain so throughout their life cycle, in which examples of hermaphroditism or spontaneous sex reversal are very rare, and these exceptional intersex fish are also termed abnormal intersexes [2]. More than 100 historical examples of abnormal hermaphroditism have been reported, but its causes remain unknown [6]. Environmental effects or genetic mutations may lead to abnormal hermaphroditism in gonochoristic fish [7]. Endocrine imbalances or abnormal hormone levels may result in intersexual conditions in gonochoristic fish [1]. In addition, xenoestrogenic compounds or chemical exposure in highly to moderately contaminated areas has also been reported to generate intersex gonads in gonochoristic fish species [8,9]. These intersexes provide opportunities to explore sex differentiation and reproductive or evolutionary mechanisms [10] and can also be used to construct inbred lines, which are significant for commercial breeding [11].

During the sexual maturation of fish, dramatic shifts in steroid biosynthetic processes occur, including changes in reproductive hormone levels and gene expression. During oocyte maturation, a shift in steroidogenesis from E2 (17β-estradiol) to 17α, 20β DHP (17α, 20β-dihydroxyprogesterone) was reported to be a crucial step, and steroids T (testosterone) and 11-KT (11-ketotestosterone, a potent androgen in fishes) play a critical role in sperm maturation and testicular development [12]. Fish synthesize luteinizing hormone (LH) and follicle-stimulating hormone (FSH) from the anterior pituitary controlled by the hypothalamic gonadotropin-releasing hormone (GnRH), to regulate early gametogenesis, steroidogenesis, and ovulation/spermiation [13]. Androgens and estrogens substantially influence the formation of primary and secondary sexual characteristics. Generally, estrogens are responsible for ovarian differentiation and feminization, whereas androgens are essential for sex determination, testis organization, male-typical spermatogonial stem cells (SSCs), and fertility [14,15]. However, the existence of synchronous intersexes (which allow simultaneous maturation of both male and female germ tissues simultaneously) reveals that exposure to changing levels of circulating hormones is probably only partly involved in the sex differentiation process. Local paracrine influences and/or differences in the reception of hormonal signals may also play significant roles [6].

Flounder is a major edible fish species worldwide; *Paralichthys olivaceus* is an important gonochoristic fish in Asia that exhibits stable sexual development throughout its life span under normal cultivation [16]. In addition, the sex determination of Japanese flounder has been documented [17], and sex-reversed males have been reported in artificial production [18]. Furthermore, treatment with a high water temperature, an aromatase inhibitor (fadrozole), or 17α-methyltestosterone during sex differentiation could induce sex reversal by suppressing *P450arom* gene expression [19]; no examples of synchronously mature intersex flounder during the process were reported.

This study is the first to discover a rare intersex member in *P. olivaceus* captured in the sea. To broaden understanding of the reproductive regulation mechanism, we fully investigated the anatomy, hormone endocrinology, molecular biology, and physiology of the synchronously mature intersex member and mature females and males in *P. olivaceus*.

## 2. Materials and Methods

### 2.1. Animals

We captured wild *P. olivaceus* in the Bohai Sea annually. In 2022, 50 *P. olivaceus* members were captured from the Qinhuangdao section of the Bohai Sea and cultured at the Beidaihe Central Experimental Station. All the captured fish were maintained under identical environmental conditions, including the density, water flow rate, and feed type and level. Briefly, *P. olivaceus* members were tagged with electronic markers by subcutaneous injection in 100 m^3^ concrete tanks and exposed to changing light conditions at varying temperatures with the seasons. In January, the parents were induced to reach sexual maturity by gradual heating and increased light exposure. After 50 days, light exposure was extended to 18 h, the water temperature reached 12.5 °C, and spawning began. In March 2023, one of the 50 wild-captured flounder was found to be a synchronously mature intersex flounder through artificial egg squeezing and sperm collection.

### 2.2. Dissection

Before dissection, the sperm and eggs of the intersex Japanese flounder, three female Japanese flounder, and three male Japanese flounder were artificially extracted by abdominal compression. The diluted sperm of the flounder was thoroughly mixed with the eggs, allowed to stand at room temperature for 2 min, and then gently mixed with seawater at 16 °C for 5 min. The resulting embryos were cultured until hatching. Next, *P. olivaceus* was anesthetized using MS-222 (200 mg/L). All dissections were performed on ice.

### 2.3. Histology

Parts of the ovaries and testes were fixed in Bouin’s solution (75% supersaturated picric acid solution, 25% formaldehyde, and 5% glacial acetic acid) for 24 h, and then stored in 70% ethanol. For a histological analysis, the samples were dehydrated using series of graded alcohol and xylene solutions and then embedded in paraffin wax. Serial sections (5–8 μm) were stained with hematoxylin and eosin (HE). The preparations were observed under a Leica DM4000 microscope (Leica Microsystems, Wetzlar, Germany), and photographs were taken using a Leica DFC490 CCD digital camera (Leica Microsystems, Wetzlar, Germany).

### 2.4. DNA Isolation and Whole-Genome Resequencing

The ovaries and testes of the intersex flounder were preserved at −80 °C for the purpose of DNA extraction. In addition, control groups comprising 30 *P. olivaceus* fins procured from the cultivated population in Hebei province were frozen at −80 °C for DNA extraction. Genomic DNA was extracted from tissues using a genomic DNA extraction kit (DP316; Tiangen, Beijing, China) in accordance with the manufacturer’s instructions. DNA quality was checked using a Qubit^TM^ 4 Fluorometer (Q33238, Thermo Fisher Scientific, Singapore) and 1% agarose gel electrophoresis. The whole-genome resequencing library was prepared according to the protocol at Novogene Co., Ltd. (Tianjin, China). Briefly, 200 ng of genomic DNA was fragmented to a size of 350 bp by sonication. The DNA fragments were end-polished, A-tailed, and ligated with a full-length adapter for Illumina sequencing, followed by further PCR amplification. The PCR products were purified using the AMPure XP system (Beverly, MA, USA). Subsequently, library quality was assessed on the Agilent 5400 system (Agilent, Santa Clara, CA, USA) and quantified by QPCR (1.5 nM). The qualified libraries were pooled and sequenced on the Illumina HiSeq X Ten platform (Novogene, Beijing, China) using 150 bp paired-end reads.

The raw data obtained by sequencing were filtered out of the low-quality sequencing data in the original sequencing data using Fastp v0.20.0 software [20]. The filtering standards were as follows: 1. Reads containing adapter sequences were filtered. 2. Reads with more than 10% of the total number of bases in a single-end read that could not be determined were excluded. 3. Reads with low-quality bases exceeding 50% of their length were filtered out. Effective, high-quality sequencing data were aligned to the reference genome using BWA (0.7.17-r1188) [21] software, and the alignment results were used to remove duplicates using GATK 4 software [22]. SNPs were identified using the HaplotypeCaller module of GATK software with a group calling strategy. The gvcf files were merged and converted to VCF files using the CombineGVCFs and GenotypeGVCFs modules. Finally, the SNPs were hard-filtered using the VariantFiltration module with the filter expression “QD < 2.0||MQ < 40.0||FS > 60.0||SOR > 3.0||MQRankSum < −12.5||ReadPosRankSum < −8.0”.

Based on the filtered SNPs’ variation file, the PLINK software (1.90 β 7) [23] was used to calculate the observed heterozygosity (Ho), expected heterozygosity (He), and homozygosity (Hom) of different population variation sites. Nucleotide diversity (PI) was calculated using VCFtools [24], with the sliding-window window size set to 50 Kb and the step size of the window step set to 10 Kb.

### 2.5. Steroid Hormone Measurement Using Enzyme-Linked Immunosorbent Assay (ELISA)

During breeding season, blood was collected by caudal venipuncture from the intersex (treatment group), 30 female (control group), and 30 male (control group) Japanese flounder. The plasma concentrations of 17β-estradiol (E2), 11-ketotestosterone (11-KT), 17α, 20β-dihydroxyprogesterone (17α, 20β DHP), luteinizing hormone (LH), follicle-stimulating hormone (FSH), testosterone (T), and 21-hydroxylase (21-OH) activity were quantified using ELISA kits (Hu Ding Biological Technology Co., Ltd., Shanghai, China). The assays were performed according to the manufacturer’s instructions, involving the sequential addition of samples, standards, and HRP-conjugated detection antibodies to the wells pre-coated with the antibody. Following incubation and thorough washing, the TMB substrate was added, leading to color development, which was subsequently stopped to halt the reaction. Color intensity was positively correlated with the concentration of the samples. The absorbance (OD values) was measured at 450 nm using a microplate reader (Tecan Inffnite 200 Pro, Männedorf, Switzerland) to determine the concentration of the samples.

### 2.6. Quantitative Real-Time PCR (qRT-PCR)

During breeding season, the ovaries and testes of three female and three male Japanese flounder were cryopreserved at a temperature of −80 °C to extract RNA. The ovarian and testicular tissues of the intersex flounder were also cryopreserved at a temperature of −80 °C to extract RNA. Total RNA was extracted using the TRIzol reagent (Invitrogen, Carlsbad, CA, USA) according to the manufacturer’s instructions. An ultra-machine spectrophotometer (P100+, Pultton, San Jose, CA, USA) was used to detect the sample concentrations. Samples with a 260/280 ratio ranging from 1.8 to 2.0 were chosen for reverse transcription. The integrity of the RNA samples was assessed by 1% liposuction gel swimming.

Primers for *cyp19a*, *amh*, and *dmrt1* were generated using the Premier 5 program. The internal reference gene utilized was *β-actin*. The primers used in this study are listed in Table 1. QRT-PCR tests were conducted in a 20 µL reaction volume comprising 1.0 µL of 100 ng/µL cDNA, 1.0 µL of each forward and reverse primer (10 µM), 10 µL of TB Green (Takara, Osaka, Japan), and 7 µL of ultrapure water. The qRT-PCR conditions were optimized with an initial denaturation at 94 °C for two minutes, followed by 40 cycles at 94 °C for 30 s and 60 °C for 20 s. Each experiment was performed in triplicate under cycling conditions, and relative expression levels were determined utilizing the 2^−ΔΔCt^ method.

### 2.7. Measuring Experimental Indicators

Reproductive performance indexes used in this study were fertilization, hatching, and deformity rates computed as follows:Fertilization rate (%) = number of gastrulas/total number of eggs × 100.
Hatching rate (%) = number of newly hatched larvae/number of gastrulas × 100.
Deformity rate (%) = number of abnormal larvae/numbers of newly hatched larvae × 100.

### 2.8. Statistics

ELISA data, qRT-PCR data, and reproductive performance index data were analyzed using a one-way analysis of variance (ANOVA) using software SPSS 13.0 with the significance set at *p* < 0.05.

## 3. Results

### 3.1. Physical Description of Intersex Flounder

In 2022, 50 *P. olivaceus* members were obtained from the Bohai Sea and cultured at the Beidaihe Central Experimental Station. And in 2023, we found that one was intersex, weighing 3 kg and resembling a female fish with an enlarged soft abdomen. When squeezing the abdomen, no eggs were squeezed out; surprisingly, only a small amount (0.01 mL) of semen was squeezed out. After dissecting the abdominal cavity in female and male *P. olivaceus*, the upper abdominal muscles were separated from each organ (Figure 1A,C). In the intersex flounder, the upper ovarian membrane was closely connected to the abdominal muscles and internal organs (Figure 1B); thus, the membrane and organs cannot be separated through physical methods. In female *P. olivaceus*, the mature eggs filled the entire ovarian cavity (Figure 1A), and the ovary appeared to be orange, and had rich vasculature and a transparent ovarian membrane (Figure 1D). Meanwhile, in the intersex flounder, the mature eggs filled their entire abdomen (Figure 1B), and the lower ovary was small and had few blood capillaries, closely connected to the testis (Figure 1E). The exterior of the testis was similar to that of male *P. olivaceus* (Figure 1F).

### 3.2. Histological Characteristics of the Intersex Flounder

The histological results showed that the ovaries of the intersex flounder were similar to those of the control group (Figure 1G,I). Differently, the testis of the intersex flounder contained a large number of spermatocytes at various stages, and the number of mature spermatozoa was less (Figure 1H,J). In the control group, the seminal vesicles were full of mature spermatozoa.

### 3.3. The Intersex Flounder Produced Both Eggs and Sperm

Sperm for the intersex were used to fertilize eggs for an unrelated control female, and the results showed normal larval development until day nine (Figure 1K). Visible non-fertilized eggs were collected by dissecting the ovary, and these eggs were used to fertilize sperm from the intersex testis and normal sperm from an unrelated control male, which also produced normal larvae (Figure 1L,M). There were no significant differences in the fertilization, hatching, or deformity rates among the three groups (Figure 1N).

### 3.4. The Intersex Flounder Had Unique Endocrine Characteristics

Hormone results indicated that LH, FSH, 17α, 20β DHP, and E2 levels in sexually mature female *P. olivaceus* were significantly higher compared to mature males, while 11-KT and T levels in sexually mature male *P. olivaceus* were significantly higher compared to mature females (Figure 2A). Interestingly, the levels of the six reproduction-related hormones in the intersex flounder were intermediate, between those in mature female and male fish (Figure 2A), displaying different endocrine characteristics from those of female and male *P. olivaceus*.

### 3.5. The Intersex Flounder Had Unique Gene Expression Profiles

Sex-related gene expression profiling was conducted on the ovary and testis of the intersex flounder. The expression of *cyp19* (P450arom gene, LOC109624967) in the female ovaries was significantly higher than that in the testes of male and intersex flounder, and the *dmrt1* and *amh* in the testes and ovaries of the intersex flounder were significantly higher than in the testes of male and female ovaries (Figure 2B). And, *dmrt1* is expressed at similar levels in intersex ovaries and testes, but *amh* is expressed higher in ovaries than in intersex testes (Figure 2B).

### 3.6. Resequencing and Activity Detection Revealed cyp21a (Steroid 21-Hydroxylase-like Gene, LOC109633161) Gene Mutation in the Intersex Flounder

To investigate the cause of the synchronously mature hermaphroditic *P. olivaceus*, we conducted resequencing analyses. Resequencing results showed that the heterozygosity of the intersex flounder was 0.632 (Figure 2C), almost one and a half times as high as that of the cultivated population (0.424). Surprisingly, in the intersex flounder, there were 28 SNPs in the LOC109633161 (*cyp21a*) gene (Appendix A), and 21-hydroxylase activity decreased by approximately 20% compared with that of mature female and male *P. olivaceus* (Figure 2D).

## 4. Discussion

*P. olivaceus* is a gonochoristic fish. In a natural sea area, males mature earlier than females [25]. Herein, a rare and synchronously mature intersex member of *P. olivaceus* in the Bohai Sea was discovered, and its reproductive characteristics, including detailed anatomy, hormone endocrinology, molecular biology, and physiology, were described, providing important data for revealing the regulatory mechanism of reproduction in *P. olivaceus*.

Many teleosts are gonochorists, who develop only as one sex and remain the same sex throughout their life spans, just like *P. olivaceus*. Meanwhile, some gonochoristic fish species undergo a period in which all gonads are initially intersexual before differentiation into either testes or ovaries, and mature as only one sex [6]. As a gonochoristic fish, *P. olivaceus* remains the same sex throughout its life span, so the synchronously mature intersex *P. olivaceus* is rare. Intersexes are identified at a very low frequency in wild populations [26], and the causes of the rare hermaphroditism, such as environmental effects (temperature or xenobiotics) or variances in sex-determination physiology, are unknown [6]. *Tachysurus fulvidraco*, a gonochoristic fish, was treated with 17α-methyltestosterone, letrozole, and a high temperature, and a certain proportion of intersexes appeared; differently, these intersexes had normally developed ovaries and testes [15]. In addition, the intersex condition has been reported in both freshwater and marine fish related to chemical exposure in highly to moderately contaminated areas [9], and these intersex males display a lower reproductive capacity than that of normal males [27], which is similar to our results. The intersex *P. olivaceus* had a lower reproductive capacity, with less mature spermatozoa in testis. Furthermore, heterozygosity of the intersex *P. olivaceus* (0.632) was one and a half times as high as that of the cultivated population (0.424), close to the wild *P. olivaceus* in the Qinhuangdao offshore area (0.64) [28]; it is speculated that synchronously mature intersex *P. olivaceus* may be related to chemical exposure during the juvenile fish stage.

During the sexual maturation of fish, dramatic shifts in steroid biosynthetic activity occur. In females, the production of T and estradiol by the ovary are reduced, and 17α, 20β DHP is dramatically enhanced. Meanwhile, in males, androgen production remains high throughout sexual maturation [29]. In *P. olivaceus*, E2 levels reach their peak when the females mature and ovulate, and T levels reach their peak when the males mature [30]. In addition, E2 was found at much higher levels in females than males [31]. It was consistent with our results; females had higher 17α, 20β-DHP and E2 levels, and males had higher T and 11-KT levels. As an inducer of the ovary, E2 plays a key role in ovarian differentiation, development, and maturation; FSH promotes follicle development and maturation; LH triggers the ovulation of females; and 17α, 20β-DHP promotes final oocyte maturation [32]. Therefore, LH, FSH, 17α, 20β-DHP, and E2 play an important role in the maturation and ovulation in female fish. And T and 11-KT play an important role in the development of spermatocytes, spermatocyte formation, and spermatogenesis [33]. The intersex *P. olivaceus* had a synchronously mature ovary and testis, so the reproductive hormone levels were intermediate between those in mature female and male fish. *Dmrt1* and *amh* are highly expressed in the testes of *P. olivaceus*, and thus *cyp19a* reaches its peak before egg laying in the ovaries [16], which correlates with the hormone levels at the corresponding developmental stage. Then, the unique expression patterns of *cyp19*, *amh*, and *dmrt1* in intersex flounder might be related to the unique hormonal characteristics.

*CYP21A2*, a cytochrome P450 enzyme situated within the endoplasmic reticulum, plays a pivotal role in humans by facilitating the conversion of 17-hydroxyprogesterone (17OHP) into 11-deoxycortisol, as well as the conversion of progesterone into 11-deoxycorticosterone [34]. *CYP21A2* deficiency in humans leads to congenital adrenal hyperplasia, manifested by corticosteroid deficiency, secondary hyper-androgens, and even pseudohermaphroditism in women [35,36]. In our results, the intersex *P. olivaceus* had 28 SNPs in the *cyp21a* gene and the activity of 21-hydroxylase was reduced by about 20%, which is similar to the clinical symptoms of congenital adrenal hyperplasia. Thus, it is speculated that the synchronously sexual maturation of intersex *P. olivaceus* may be related to the endocrine imbalance caused by the decreased activity of 21-hydroxylase. Therefore, the mechanism of the synchronously sexual maturation of intersex *P. olivaceus* can be further investigated through the knockout of the *cyp21a* gene of *P. olivaceus*.

## 5. Conclusions

In summary, we present the first report of intersex *P. olivaceus*, including detailing the anatomy, hormone endocrinology, molecular biology, and physiology to reveal the relevance of hormone levels in gonadal development. It is important for the studying of the reproductive endocrine regulatory mechanism of *P. olivaceus* (Figure 3).

## Figures and Tables

**Figure 1 animals-14-02948-f001:**
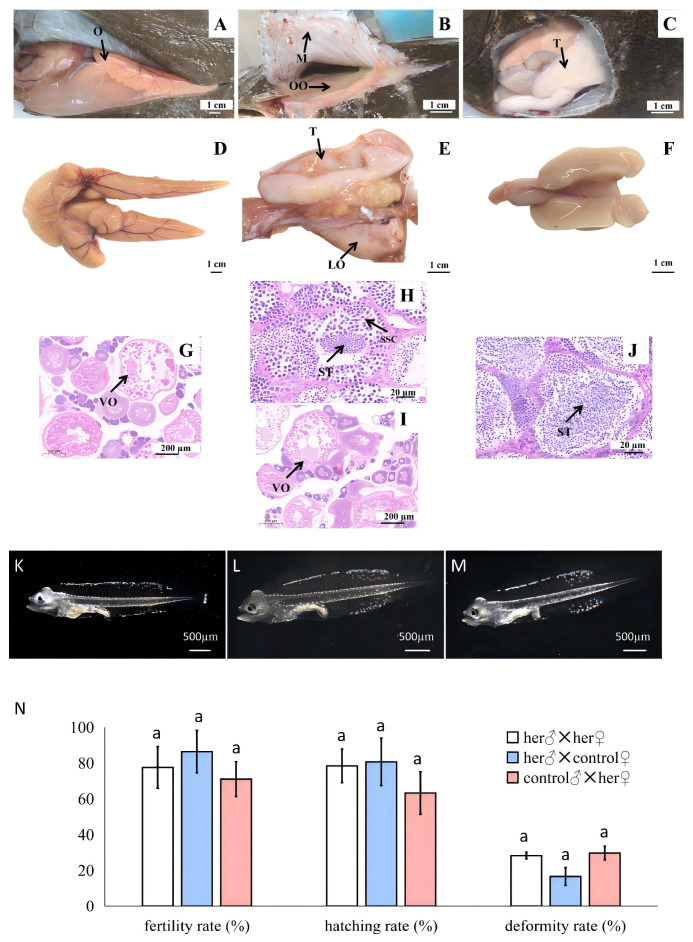
Anatomical and histological characteristics of intersex gonads in Japanese flounder. (**A**–**C**) Anatomical characteristics of ovary in female, intersex, and male Japanese flounder, respectively. (**D**–**F**) Morphological characteristics of ovary in female, intersex, and male Japanese flounder, respectively. (**G**–**I**) Histological characteristics of ovary in female and intersex Japanese flounder, respectively. (**H**–**J**) Histological characteristics of testis in intersex and male Japanese flounder. (**K**) Normal development through day 9 from crossing intersex sperm with control unrelated female eggs. (**L**) Normal development of larvae using intersex self-fertilization, and sperm from testis and ovary of same animal were crossed, yielding normal development through day 9. (**M**) Normal development through day 9 from crossing intersex eggs with unrelated control male sperm. (**N**) Fertility, hatching, and deformity rate comparison among three crosses’ groups. O indicates ovary, M indicates membrane of upper ovary, OO indicates oocytes, T indicates testis, LO indicates lower ovary, ⅤO indicates phase Ⅴ oocytes, SSC indicates secondary spermatocytes, and ST indicates spermatoon. Same letter means difference was not significant.

**Figure 2 animals-14-02948-f002:**
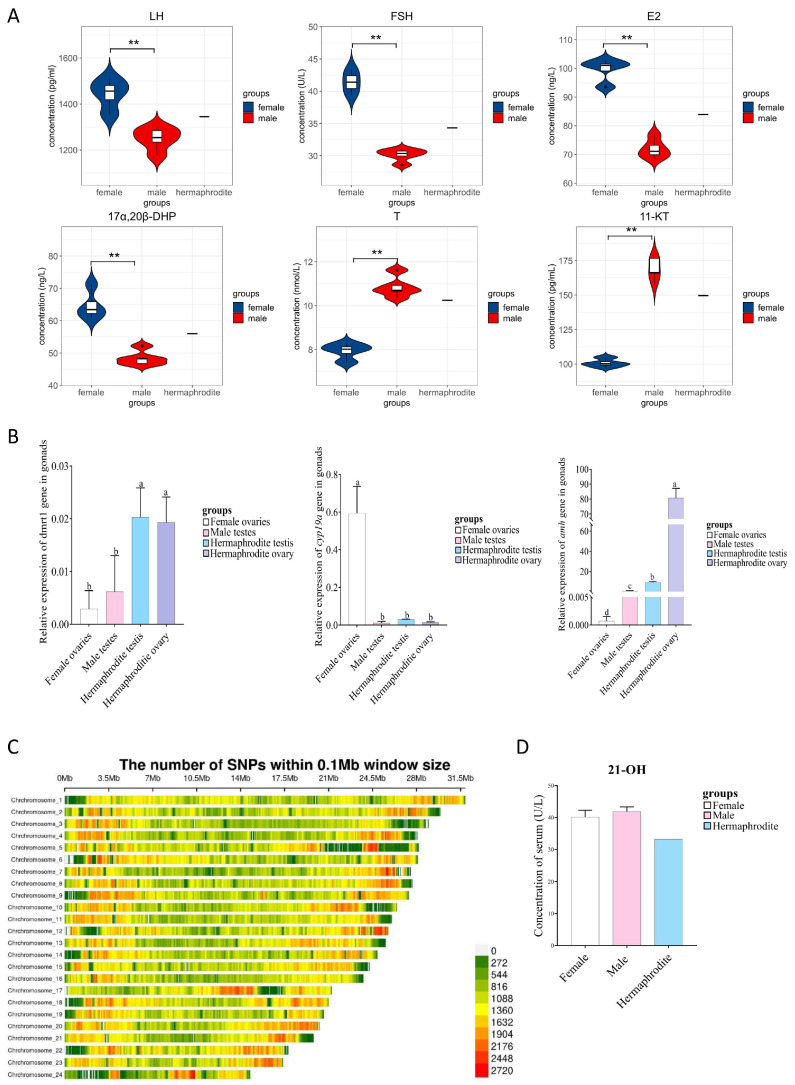
Molecular and physiological characteristics of intersex gonads in Japanese flounder. (**A**) Comparison of hormones’ level. (**B**) Differential gene expression profiling. (**C**) Density plot of SNPs. (**D**) 21-hydroxylase activity comparison. Different letters means difference was not significant. ** indicates *p* ˂ 0.01.

**Figure 3 animals-14-02948-f003:**
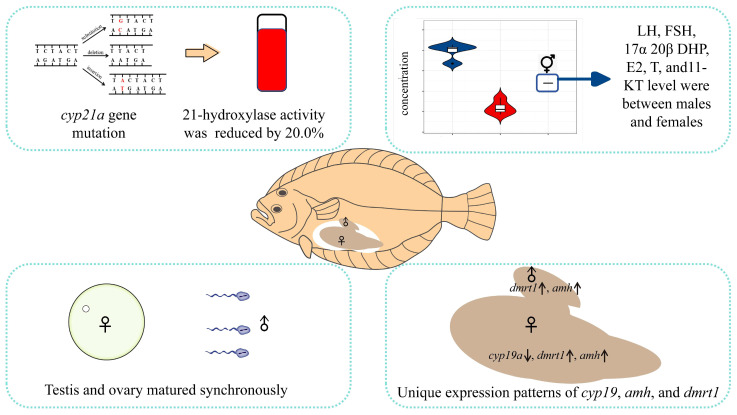
The *Paralichthys olivaceus* is a gonochoristic fish. We report an unusual case of hermaphroditism in the Japanese flounder, which may be related to *cyp21a* gene mutation by xenoestrogenic compounds or other chemical exposure. Compared with that of wild flounder, the activity of 21-hydroxylase was reduced by approximately 20.0%; the levels of several reproduction-related hormones in the intersex flounder were intermediate, between those in female and male fish; and expressions of *cyp19a*, *amh*, and *dmrt1* also differed, and both the testis and ovary matured synchronously. 
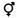
 indicates the intersex, ♂ indicates male, ♀ indicates female, ↑ indicates up-regulated, ↓ indicates down-regulated.

**Table 1 animals-14-02948-t001:** Primers used in this study.

Primers	Sequence (5′–3′)
*cyp19a*-F	TTGGGAGCAAGCAGGGACT
*cyp19a*-R	TGGTGAAATGGGTGCGTAT
*amh*-F	TATCGCTGGGCTTGTCCTC
*amh*-R	CCCCTTATCACCTCCATCAT
*dmrt1*-F	GGCACCAGCACAGGCATCG
*dmrt1*-R	AGCGGGAGCACTTGGGCAT
*β-actin*-F	GGAAATCGTGCGTGACATTAAG
*β-actin*-R	CCTCTGGACAACGGAACCTCT

## Data Availability

The SNP genotypes and raw sequence data for this article are available in the China National Center for Bioinformation, accessible at https://ngdc.cncb.ac.cn/gsa/search?searchTerm=CRA018591 (accessed on 27 August 2024), and are also available under accession number PRJCA029411. Otherwise, the authors state that the data necessary to confirm the conclusions presented in the article are represented fully within the article.

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
