# Peer review of "Synchronously Mature Intersex Japanese Flounder (Paralichthys olivaceus): A Rare Case"

_animals, 2024, doi:10.3390/ani14202948_

Round 1

Reviewer 1 Report

Comments and Suggestions for Authors

This paper investigates the intersex individuals that have appeared in the normally gonochoristic flounder, using anatomy, hormone endocrinology, molecular biology and physiology. This paper is interesting in that it attempts to clarify the reproductive regulation mechanisms of the flounder by measuring sex differentiation-related genes, sex hormones, etc. in individuals with simultaneously fertilisable eggs and sperm, and comparing them with normal females and males. However, this study lacks the necessary information to replicate and validate each experiment.

My specific comments are listed below.

Major comments

1. L114-116

Is the individual from which the fins were collected different from the individual described in 2.2? If so, information on the individuals from which fins were collected should be provided in 2.2 and the method used when the fins were collected in 2.3.

Also, why was DNA collected from the ovaries and testes in intersex individuals while DNA was collected from the fins in control individuals? Furthermore, are individuals from which fins are collected sexed from the gonads? Since the Introduction states that genetic and phenotypic sex may differ under captive conditions, is it possible that sex-changed individuals are included?

2. L182-184

The paper states that intersex was observed in only one out of 50 individuals.

In Fig. 1O, intersex appears to be indicated by hormone concentrations in only one individual. But why are there error bars in Fig. 1N and Fig. 1P for data from one intersex individual?

The authors should indicate the sample size of the data used in the graphs for each group.

Also, Fig. 1O is shown as a box-and-whisker and violin plot, rather than a bar chart showing mean ± SD. If the data notation differs for each experiment, an explanation of why the data notation differs should be provided in 2.8.

Furthermore, the analysis of variance does not determine which groups are significantly different. Please describe exactly how the statistical analysis of each set of data was carried out.

3. L242-245

dmrt1 is expressed at similar levels in intersex ovaries and testes, but amh is expressed higher in ovaries than in intersex testes. It would be better to mention this point.

Also, as there is no discussion in this paper regarding the differences in the expression of these genes in the intersex gonads compared to normal sexes, the authors should discuss the reasons for this.

Furthermore, can intersex data not be statistically analysed like ELISA results?

4. L294-308

This paragraph is a description of sex hormones in general and does not seem appropriate for discussion. It would be better to integrate the previous paragraph with this paragraph and discuss why each hormone shows intermediate values between the sexes in intersex individuals.

Minor comments

1. L92-99

Please indicate changes in water temperature and daylength conditions during the experimental period.

2. L98-99

This information should be included in the results.

3. L101-103

Were the sperm and eggs of Intersex individuals also collected before dissection? Please also indicate the fertilisation method and incubation conditions up to hatching.

4. L107-109

As 100% ethanol and paraffin do not mix, I think the common method is to replace it with an intermediate agent such as xylene before paraffin embedding.

Is the method of paraffin embedding correctly described?

5. L114-115

Please describe how testes and ovaries are preserved.

6. L127

Please provide information on the sequencing equipment used in this study.

7. L148

Please describe the blood collection method.

8. L221-222, L229-230, L238-239, L245-246, L254-258

This information should be included in the discussion.

9. L281-284

The rationale for chemical exposure being associated with intersex was not clear from this description. I think additional explanations are needed to make it clearer to the reader.

10. L292-293

What specific studies do the authors think are needed to elucidate the mechanisms?

Author Response

Dear editor,

Thanks for your works of our manuscript.

We have read the comments carefully and tried our best to modify our manuscript by the requested points.

We hope that you can be satisfied with it and give us more advices to improve our paper.

There are the comments and our answer as follows.

Comments and Suggestions for Authors

This paper investigates the intersex individuals that have appeared in the normally gonochoristic flounder, using anatomy, hormone endocrinology, molecular biology and physiology. This paper is interesting in that it attempts to clarify the reproductive regulation mechanisms of the flounder by measuring sex differentiation-related genes, sex hormones, etc. in individuals with simultaneously fertilisable eggs and sperm, and comparing them with normal females and males. However, this study lacks the necessary information to replicate and validate each experiment.

 My specific comments are listed below.

Major comments

  1. L114-116

Is the individual from which the fins were collected different from the individual described in 2.2? If so, information on the individuals from which fins were collected should be provided in 2.2 and the method used when the fins were collected in 2.3.

Answer: Thanks for your suggestion. The individuals described in 2.2 and 2.3 were same. According to your advice, we revised 2.2 and 2.3 as follows:

”2.2. Animals

We captured wild P. olivaceus in the Bohai Sea annually. In 2022, 50 P. olivaceus were captured from the Qinhuangdao section of Bohai Sea and cultured at the Beidaihe Central Experimental Station. All the captured fish were maintained under identical environmental conditions, including density, water flow rate, and feed type and level. Briefly, P. olivaceus were tagged with electronic markers by subcutaneous injection in 100 m3 concrete tanks and exposed to changing light conditions at varying temperatures with the seasons. In January, the parents were induced to reach sexual maturity by gradual heating and increased light exposure. After 50 days, light exposure was extended to 18 hours, the water temperature reached 12.5 ℃ and spawning began. In March 2023, one of the 50 wild-captured flounders was found to be a synchronously mature intersex through artificial egg squeezing and sperm collection.

2.3. Dissection

Before dissection, the sperm and eggs of the intersex Japanese flounder, three female and three male Japanese flounder were artificially extracted by abdominal compression. The diluted sperm of the flounder was thoroughly mixed with the eggs, allowed to stand at room temperature for 2 minutes, then gently mixed with seawater at 16°C for 5 minutes. The resulting embryos were cultured until hatching. Next, P. olivaceus was anesthetized using MS-222 (200 mg/L). All dissections were performed on ice.“ 

Also, why was DNA collected from the ovaries and testes in intersex individuals while DNA was collected from the fins in control individuals? Furthermore, are individuals from which fins are collected sexed from the gonads? Since the Introduction states that genetic and phenotypic sex may differ under captive conditions, is it possible that sex-changed individuals are included?

Answer: Thanks for your suggestion. It was generally believed that the DNA information among all tissues of a individual was consistent, in order to reduce the numbers of experimental animals killed during sampling process,we had not collected the gonads of the control group.

Because the purpose of the experiment was to detect the genetic diversity of intersex and whether there were gene mutations, 30 individuals as control group were randomly selected from cultured populations, and we had not distinguish between males and females in the control group populations.

As for the captive conditions mentioned in the introduction, genetic and phenotypic sex may be different, so sex-changed individuals may be included, and this captive condition is generally induced by artificial intervention of pseudo-male and pseudo-female fish, including intervention of external conditions such as hormones and temperature. The control group in the study were the normal cultured populations, without manual intervention, the possibility of intersex was very small.

  1. L182-184

The paper states that intersex was observed in only one out of 50 individuals.

In Fig. 1O, intersex appears to be indicated by hormone concentrations in only one individual. But why are there error bars in Fig. 1N and Fig. 1P for data from one intersex individual?

Answer: Thank you for your suggestion. Each experiment was repeated three times, and statistical data of fertilization rate, hatching rate and malformation rate would appear for three times. The difference analysis of these data would show error lines in Fig. 1N. Similarly, three experiments of gene expression were conducted, so there was error lines in Fig. 1P.

The authors should indicate the sample size of the data used in the graphs for each group.

Answer: According to your suggestion, we added the sample size of the data used in the graphs for each group in the Materials and methods section as follows:

”2.3. Dissection

Before dissection, the sperm and eggs of the intersex Japanese flounder, three female and three male Japanese flounder were artificially extracted by abdominal compression. The diluted sperm of the flounder was thoroughly mixed with the eggs, allowed to stand at room temperature for 2 minutes, then gently mixed with seawater at 16°C for 5 minutes. The resulting embryos were cultured until hatching. Next, P. olivaceus was anesthetized using MS-222 (200 mg/L). All dissections were performed on ice.

2.6. Steroid hormone measurement using enzyme-linked immunosorbent assay (ELISA)

During breeding season, blood were collected by caudal venipuncture from the intersex (treatment group), 30 female (control group) and 30 male (control group) Japanese flounders. The plasma concentrations of 17β-estradiol (E2), 11-ketotestosterone (11-KT), 17α, 20β-dihydroxyprogesterone (17α, 20β DHP), luteinizing hormone (LH), follicle-stimulating hormone (FSH), testosterone (T), and 21-hydroxylase (21-OH) activity were quantified using ELISA kits (Hu Ding Biological Technology Co., Ltd., Shanghai, China).

2.7. Quantitative real-time PCR (qRT-PCR)

During breeding season, the ovaries and testes of three female and three male Japanese flounders were cryopreserved at a temperature of -80℃ to extract RNA. The ovarian and testicular tissues of the intersex were also cryopreserved at a temperature of -80℃ to extract RNA. “ 

Also, Fig. 1O is shown as a box-and-whisker and violin plot, rather than a bar chart showing mean ± SD. If the data notation differs for each experiment, an explanation of why the data notation differs should be provided in 2.8.

Answer: Thank you for your suggestion. Fig. 1O is shown as a violin plot, which clearly showing hormone levels between the control group and intersex.

Furthermore, the analysis of variance does not determine which groups are significantly different. Please describe exactly how the statistical analysis of each set of data was carried out.

Answer: According to your suggestion, we added the statistical analysis in all experiments as follows:

”2.9. Statistics

ELISA data, qRT-PCR data and reproductive performance index data were analyzed using one-way analysis of variance(ANOVA) using software SPSS 13.0 with the significance set at P < 0.05.”

  1. L242-245

dmrt1 is expressed at similar levels in intersex ovaries and testes, but amh is expressed higher in ovaries than in intersex testes. It would be better to mention this point.

Answer: According to your suggestion, we added the sentence in L242-245 as follows:”dmrt1 is expressed at similar levels in intersex ovaries and testes, but amh is expressed higher in ovaries than in intersex testes”.

Also, as there is no discussion in this paper regarding the differences in the expression of these genes in the intersex gonads compared to normal sexes, the authors should discuss the reasons for this.

Furthermore, can intersex data not be statistically analysed like ELISA results?

Answer: According to your suggestion, we added the discussion in this paper regarding the differences in the expression of these genes in the intersex gonads compared to normal sexes as follows:”Dmrt1 and amh are highly expressed in the testes of P. olivaceus, thus cyp19a reaches its peak before egg laying in the ovaries [16], which correlated with the hormone levels at the corresponding developmental stage. Then, the unique expression patterns of cyp19, amh, and dmrt1 in intersex might be related to the unique hormonal characteristics.

  1. L294-308

This paragraph is a description of sex hormones in general and does not seem appropriate for discussion. It would be better to integrate the previous paragraph with this paragraph and discuss why each hormone shows intermediate values between the sexes in intersex individuals.

Answer: According to your suggestion, we added the discussion in this paper as follows:”During sexual maturation of fish, dramatic shifts in steroid biosynthetic activity occur. In females, the production of T and estradiol by the ovary were reduced, and 17α, 20β DHP was dramatically enhanced. While in males, androgen production remained high throughout sexual maturation [29]. In P. olivaceus, E2 levels reached peak when the females matured and ovulated, and T levels reached peak when the males matured [30]. In addition, E2 was found at much higher levels in females than males [31]. It was consistent with our results, females had higher 17α,20β-DHP and E2 level, and males had higher T and 11-KT level. As an inducer of the ovary, E2 played a key role in ovarian differentiation, development and maturation, FSH promoted follicle development and maturation, LH triggered ovulation of females, 17α,20β-DHP promoted final oocyte maturation [32]. Therefore, LH, FSH, 17α,20β-DHP, and E2 played important role in the maturation and ovulation in females of fish. And T and 11-KT played an important role in the development of spermatocytes, spermatocyte formation and spermatogenesis [33]. The intersex P. olivaceus had synchronously mature ovary and testis, so the reproductive hormones level were intermediate between those in mature female and male fish. Dmrt1 and amh are highly expressed in the testes of P. olivaceus, thus cyp19a reaches its peak before egg laying in the ovaries [16], which correlated with the hormone levels at the corresponding developmental stage. Then, the unique expression patterns of cyp19, amh, and dmrt1 in intersex might be related to the unique hormonal characteristics.

CYP21A2, a cytochrome P450 enzyme situated within the endoplasmic reticulum, plays a pivotal role in humans by facilitating the conversion of 17-hydroxyprogesterone (17OHP) into 11-deoxycortisol, as well as the conversion of progesterone into 11-deoxycorticosterone [34]. CYP21A2 deficiency in humans leads to congenital adrenal hyperplasia, manifested by corticosteroid deficiency, secondary hyper-androgen, and even pseudohermaphroditism in women [35, 36]. In our results, the intersex P. olivaceus had 28 SNPs in cyp21a gene and the activity of 21 hydroxylase was reduced by about 20%, which as similar to the clinical symptoms of congenital adrenal hyperplasia. Thus, It is speculated that the synchronously sexual maturation of intersex P. olivaceus may be related to the endocrine imbalance caused by the decreased activity of 21 hydroxylase. Therefore, the mechanism of synchronously sexual maturation of intersex in P. olivaceus can to be further investigated through knockout of cyp21a gene of P. olivaceus.”

Minor comments

  1. L92-99

Please indicate changes in water temperature and daylength conditions during the experimental period.

Answer: Thank you for your suggestion. During non-breeding season, the cultured water temperature changed with the temperature in Bohai sea, and light varied with the seasons. Before pre-breeding, when the light reached 18h and the water temperature reached 12.5 ℃, the breeding begins, which has been described in L92-99.

  1. L98-99

This information should be included in the results.

Answer: According to your suggestion, we moved the section of L98-99 to the results.

  1. L101-103

Were the sperm and eggs of Intersex individuals also collected before dissection? Please also indicate the fertilisation method and incubation conditions up to hatching.

Answer: According to your suggestion, we added the methods in 2.3 as follows:

 ”2.3. Dissection

Before dissection, the sperm and eggs of the intersex Japanese flounder, three female and three male Japanese flounder were artificially extracted by abdominal compression. The diluted sperm of the flounder was thoroughly mixed with the eggs, allowed to stand at room temperature for 2 minutes, then gently mixed with seawater at 16°C for 5 minutes. The resulting embryos were cultured until hatching. Next, P. olivaceus was anesthetized using MS-222 (200 mg/L). All dissections were performed on ice.”

  1. L107-109

As 100% ethanol and paraffin do not mix, I think the common method is to replace it with an intermediate agent such as xylene before paraffin embedding.

Is the method of paraffin embedding correctly described?

Answer: According to your suggestion, we revised the methods in L107-109 as follows: ”For histological analysis, the samples were dehydrated using series of graded alcohol and xylene solutions and then embedded in paraffin wax. ”

  1. L114-115

Please describe how testes and ovaries are preserved.

Answer: According to your suggestion, we revised the methods in L114-115 as follows: ”The ovaries and testes of the intersex were preserved at -80℃ for the purpose of DNA extraction. ”

  1. L127

Please provide information on the sequencing equipment used in this study.

Answer: According to your suggestion, we added the information on the sequencing equipment used in this study in L127 as follows: ”The qualified libraries were pooled and sequenced on the Illumina HiSeq X Ten platform (Novogene, Beijing, China) using 150-bp paired-end reads.”

  1. L148

Please describe the blood collection method.

Answer: According to your suggestion, we added the blood collection method in L148 as follows: ”During breeding season, blood were collected by caudal venipuncture from the intersex (treatment group), 30 female (control group) and 30 male (control group) Japanese flounders. ”

  1. L221-222, L229-230, L238-239, L245-246, L254-258

This information should be included in the discussion.

Answer: According to your suggestion, we moved L221-222, L229-230, L238-239, L245-246, L254-258 to the discussion.

  1. L281-284

The rationale for chemical exposure being associated with intersex was not clear from this description. I think additional explanations are needed to make it clearer to the reader.

Answer: Thank you for your suggestion. The causes of the rare hermaphroditism, such as environmental effects (temperature or xenobiotics) or variances in sex-determination physiology, were unknown. We have not found any relevant reports yet.

  1. L292-293

What specific studies do the authors think are needed to elucidate the mechanisms?

Answer: According to your suggestion, we added the discussion in L292-293 as follows:”Therefore, the mechanism of synchronously sexual maturation of intersex in P. olivaceus can to be further investigated through knockout of cyp21a gene of P. olivaceus.”

Reviewer 2 Report

Comments and Suggestions for Authors

MS is very interesting but needs improvement. My comments are below:

Simple Summary

Line 19: Please changes “Our” on “In this study”

Abstract

Line 28: Please list in parentheses which hormones had intermediate values.

Line 29: Please explain the SNP abbreviation in brackets.

Line 34: I noticed that the authors provided only four keywords, three of which are the same as the MS title. Repeated keywords should be replaced with new ones for example: reproduction; gonadal development; hydroxylase; expressions; hormonal endocrinology

Introduction

Line 81:  Please remove the word "our".

Materials and methods

Line 90: Please provide a more precise location where the fish was caught.

Line 90: How were the fish caught and transported to the laboratory?

Line 93: Please provide details of the electronic markers used to mark the fish.

Line 93: Please indicate the density at which the fish were kept?

Line 93: What was the water flow rate?

Line 93: What feed were the fish fed? How many times a day? And what was the feed ration used?

Line 98: How was it determined that one fish is a synchrosexually mature intersex? After the dissection was performed?

Line 114: How were the ovaries and testes stored for DNA extraction? Was some solution used to store the gonads? What kind?

Line 175: Please change the name of point 2.8 to: "29. Statistical analysis".

Line 176 – 181: Please move to new section 2.9. Measuring experimental indicators.

Line 182: Statistics should be described in detail and what tests were used to analyze the data.

Results

Line 201: Figure 1 is invisible, which causes MS to lose quality. I suggest splitting Figure 1 into at least 6 - 7 figures. Figure 1: current Figures 1A to 1J. Figure 2: only Fig. 1P (three graphs). Figure 3: current Fig. 1R. Figure 4: 1O. Fig.  5: current figure 1N (3 graphs). Figure 6: current Fig. 1Q. Figure 7: current Fig.  1K - 1M.

All figures should be clear, have the appropriate font size and all axes labeled.

Discussion

The discussion presents valuable insights into the rare phenomenon of P. olivaceus, particularly focusing on its reproductive characteristics, potential causes, and hormonal regulation. However, a few key areas could be strengthened. I am missing a mention of whether and in which species intersexuality has been observed? Furthermore, I believe that MS lacks comparisons with other gonochoristic species and the role of hormones in reproduction. A broader comparative analysis of intersexual states in different species would be beneficial. This would help to place the rarity of intersexual P. olivaceus in the context of other known cases in bony fishes. The authors mention environmental and physiological factors potentially contributing to intersexual states, but there is no clear hypothesis about the mechanisms underlying intersexuality. It would be helpful to discuss in more detail whether chemical exposure or genetic factors are likely to be the main drivers. There is also a lack of specific examples or comparisons with other studies that could strengthen the discussion. More specific evidence or references to specific contaminants associated with endocrine disruption in marine environments would strengthen the link between environmental stressors and intersexuality.

Line 271: What does it mean that they are rare? That is, what percentage of the population. Please elaborate on this a bit.

Line 271: Intersexuality was rare in the wild population of P. olivaceus? Or in various wild fish populations? Please be more specific.

Author Response

Dear editor,

Thanks for your works of our manuscript.

We have read the comments carefully and tried our best to modify our manuscript by the requested points.

We hope that you can be satisfied with it and give us more advices to improve our paper.

There are the comments and our answer as follows.

Comments and Suggestions for Authors

MS is very interesting but needs improvement. My comments are below:

Simple Summary

Line 19: Please changes “Our” on “In this study”

Answer: According to your suggestion, we revised “Our” to “In this study”.

Abstract

Line 28: Please list in parentheses which hormones had intermediate values.

Answer: According to your suggestion, we added the hormones in Line 28 as follows:”( 17β-estradiol, 11-ketotestosterone, 17α, 20β-dihydroxyprogesterone, luteinizing hormone, follicle-stimulating hormone and testosterone) ”

Line 29: Please explain the SNP abbreviation in brackets.

Answer: According to your suggestion, we have added the full name of SNPs as follows:”single nucleotide polymorphisms”in Line 29.

Line 34: I noticed that the authors provided only four keywords, three of which are the same as the MS title. Repeated keywords should be replaced with new ones for example: reproduction; gonadal development; hydroxylase; expressions; hormonal endocrinology

Answer: According to your suggestion, we revised the keywords as follows:”synchronously sexual maturity; 21 hydroxylase activity; gene expressions; hormonal endocrinology”in Line 34.

Introduction

Line 81:  Please remove the word "our".

Answer: According to your suggestion, we removed the word "our" in Line 81.

Materials and methods

Line 90: Please provide a more precise location where the fish was caught.

Answer: According to your suggestion, we provided the more precise location where the fish was caught as follows:”We captured wild P. olivaceus in the Bohai Sea annually. In 2022, 50 P. olivaceus were captured from the Qinhuangdao section of Bohai Sea and cultured at the Beidaihe Central Experimental Station.”in Line 90.

Line 90: How were the fish caught and transported to the laboratory?

Answer: Thank you for your suggestion. The Japanese flounders captured with a bottom trawl were brought back to the live fish carrier, and then transferred to the Beidaihe Central Experimental Station for temporary rearing and domestication. Generally, it takes two to three months to domesticate the wild flounder to adapt to the aquaculture environment.

Line 93: Please provide details of the electronic markers used to mark the fish.

Answer: According to your suggestion, we provided the more precise as follows:”In March 2023, one of the 50 wild-captured flounders was found to be a synchronously mature intersex through artificial egg squeezing and sperm collection.”in Line 93.

Line 93: Please indicate the density at which the fish were kept?

Answer: Thank you for your suggestion. The density of fish in breeding season and non-breeding season were different, so the density was not included in the paper in Line 93.

Line 93: What was the water flow rate?

Answer: Thank you for your suggestion. The water flow rate of fish in breeding season and non-breeding season were different. Generally, we maintained a flowing water system in summer during the non-breeding season, and keeped stillwater in winter to maintain water temperature for aquaculture. During breeding season, to stimulate the gonadal development of the broodstock, we maintained a flowing water system. Therefore, there is no fixed flow rate. So the density was not included in the paper in Line 93.

Line 93: What feed were the fish fed? How many times a day? And what was the feed ration used?

Answer: Thank you for your suggestion. The feed of fish were different in different season. Generally, we feed once every one to two days in summer; in spring, autumn, and winter, we feed once a day. Moreover, the amount of feed varied with the seasons, so we did not mention the exact number of feed and the amount of feed in Line 93.

Line 98: How was it determined that one fish is a synchrosexually mature intersex? After the dissection was performed?

Answer: Thank you for your suggestion. During the breeding season each year, we conducted artificial insemination. During the process of manually stripping eggs and collecting sperm, we discovered a intersex of Japanese flounder.

Line 114: How were the ovaries and testes stored for DNA extraction? Was some solution used to store the gonads? What kind?

Answer: According to your suggestion, we added the methods as follows:”The ovaries and testes of the intersex were preserved at -80℃ for the purpose of DNA extraction. ”in Line 114.

Line 175: Please change the name of point 2.8 to: "29. Statistical analysis".

Answer: According to your suggestion, we changed the name of point 2.8 to: "29. Statistical analysis".

Line 176 – 181: Please move to new section 2.9. Measuring experimental indicators.

Answer: According to your suggestion, we moved to new section 2.9. Measuring experimental indicators.

Line 182: Statistics should be described in detail and what tests were used to analyze the data.

Answer: According to your suggestion, we added the statistics methods as follows:”ELISA data, qRT-PCR data and reproductive performance index data were analyzed using one-way analysis of variance (ANOVA) using software SPSS 13.0 with the significance set at P < 0.05 ”in Line 182.

Results

Line 201: Figure 1 is invisible, which causes MS to lose quality. I suggest splitting Figure 1 into at least 6 - 7 figures. Figure 1: current Figures 1A to 1J. Figure 2: only Fig. 1P (three graphs). Figure 3: current Fig. 1R. Figure 4: 1O. Fig. 5: current figure 1N (3 graphs). Figure 6: current Fig. 1Q. Figure 7: current Fig. 1K - 1M.

All figures should be clear, have the appropriate font size and all axes labeled.

Answer: According to your suggestion, we submitted the clear figures. Since the format we are submitting is brief report and only allows for one to two figures, so we had to combine the seven figures into one large figure.

Discussion

The discussion presents valuable insights into the rare phenomenon of P. olivaceus, particularly focusing on its reproductive characteristics, potential causes, and hormonal regulation. However, a few key areas could be strengthened. I am missing a mention of whether and in which species intersexuality has been observed? Furthermore, I believe that MS lacks comparisons with other gonochoristic species and the role of hormones in reproduction. A broader comparative analysis of intersexual states in different species would be beneficial. This would help to place the rarity of intersexual P. olivaceus in the context of other known cases in bony fishes. The authors mention environmental and physiological factors potentially contributing to intersexual states, but there is no clear hypothesis about the mechanisms underlying intersexuality. It would be helpful to discuss in more detail whether chemical exposure or genetic factors are likely to be the main drivers. There is also a lack of specific examples or comparisons with other studies that could strengthen the discussion. More specific evidence or references to specific contaminants associated with endocrine disruption in marine environments would strengthen the link between environmental stressors and intersexuality.

Answer: According to your suggestion, we revised the discussion as follows:”

  1. Discussion P. olivaceuswas a gonochoristic fish. In the natural sea area, males matured earlier than females [25]. Herein, a rare and synchronously mature intersex of P. olivaceusin the Bohai Sea was discovered, and its reproductive characteristics, including detailed anatomy, hormone endocrinology, molecular biology, and physiology, were described, providing important data for revealing the regulatory mechanism of reproduction in P. olivaceus.

Many teleosts are gonochorists, who develop only as one sex and remain the same sex throughout their life spans, just as P. olivaceus. While, some gonochoristic fish species undergo a period in which all gonads are initially intersexual before differentiation into either testes or ovaries, and mature as only one sex [6]. As a gonochoristic fish, P. olivaceus remained the same sex throughout their life spans, so the synchronously mature intersex of P. olivaceus was rare. Intersexes were identified at very low frequency in wild populations [26], and the causes of the rare hermaphroditism, such as environmental effects (temperature or xenobiotics) or variances in sex-determination physiology, were unknown [6]. Tachysurus fulvidraco, a gonochoristic fish, were treated with 17α-methyltestosterone, letrozole and high temperature, a certain proportion of intersexs might appear, differently these intersexs had normally developed ovaries and testes [15]. In addition, intersex condition has been reported in both freshwater and marine fish related to chemical exposure in highly to moderately contaminated areas [9] , and these intersex males display a lower reproductive capacity than that of normal males [27], which is similar to our results. The intersex P. olivaceus had a lower reproductive capacity, with less mature spermatozoa in testis. Futhermore, heterozygosity of the intersex P. olivaceus (0.632) was one and a half times as high as that of the cultivated population (0.424) close to the wild P. olivaceus in Qinhuangdao offshore area (0.64) [28], it is speculated that synchronously mature intersex P. olivaceus may be related to chemical exposure during juvenile fish.

During sexual maturation of fish, dramatic shifts in steroid biosynthetic activity occur. In females, the production of T and estradiol by the ovary were reduced, and 17α, 20β DHP was dramatically enhanced. While in males, androgen production remained high throughout sexual maturation [29]. In P. olivaceus, E2 levels reached peak when the females matured and ovulated, and T levels reached peak when the males matured [30]. In addition, E2 was found at much higher levels in females than males [31]. It was consistent with our results, females had higher 17α,20β-DHP and E2 level, and males had higher T and 11-KT level. As an inducer of the ovary, E2  played a key role in ovarian differentiation, development and maturation, FSH promoted follicle development and maturation, LH triggered ovulation of females, 17α,20β-DHP promoted final oocyte maturation [32]. Therefore, LH, FSH, 17α,20β-DHP, and E2 played important role in the maturation and ovulation in females of fish. And T and 11-KT played an important role in the development of spermatocytes, spermatocyte formation and spermatogenesis [33]. The intersex P. olivaceus had synchronously mature ovary and testis, so the reproductive hormones level were intermediate between those in mature female and male fish. Dmrt1 and amh are highly expressed in the testes of P. olivaceus, thus cyp19a reaches its peak before egg laying in the ovaries [16], which correlated with the hormone levels at the corresponding developmental stage. Then, the unique expression patterns of cyp19, amh, and dmrt1 in intersex might be related to the unique hormonal characteristics.

CYP21A2, a cytochrome P450 enzyme situated within the endoplasmic reticulum, plays a pivotal role in humans by facilitating the conversion of 17-hydroxyprogesterone (17OHP) into 11-deoxycortisol, as well as the conversion of progesterone into 11-deoxycorticosterone [34]. CYP21A2 deficiency in humans leads to congenital adrenal hyperplasia, manifested by corticosteroid deficiency, secondary hyper-androgen, and even pseudohermaphroditism in women [35, 36]. In our results, the intersex P. olivaceus had 28 SNPs in cyp21a gene and the activity of 21 hydroxylase was reduced by about 20%, which as similar to the clinical symptoms of congenital adrenal hyperplasia. Thus, It is speculated that the synchronously sexual maturation of intersex P. olivaceus may be related to the endocrine imbalance caused by the decreased activity of 21 hydroxylase. Therefore, the mechanism of synchronously sexual maturation of intersex in P. olivaceus can to be further investigated through knockout of cyp21a gene of P. olivaceus. ”

Line 271: What does it mean that they are rare? That is, what percentage of the population. Please elaborate on this a bit.

Answer: Thank you for your suggestion.  It was reported that intersexes were identified at very low frequency in wild populations [27], and the causes of the rare hermaphroditism, such as environmental effects (temperature or xenobiotics) or variances in sex-determination physiology, were unknown [6].  

In the wild environment, there are few reports of gonochoristic fish transitioning to hermaphroditic individuals, and such cases are often reported in the form of single instances, making this phenomenon quite rare.

Line 271: Intersexuality was rare in the wild population of P. olivaceus? Or in various wild fish populations? Please be more specific.

Answer: Thank you for your suggestion. Synchronously mature intersex in Japanese flounder has not been previously reported. And, in the past 15 year, it was the first time we had caught a synchronously mature intersex in the wild Japanese flounder. So, it is very rare. 

Round 2

Reviewer 2 Report

Comments and Suggestions for Authors

MS has been significantly improved. The only thing that bothers me is that MS itself has poor quality drawings. However, you can see them in higher resolution. I would encourage authors to divide the drawings into at least 2 parts and enlarge them. MS would definitely gain in quality then. I think MS is really good and raises very interesting issues synchronously mature intersex in Japanese flounder.

Author Response

Comments and Suggestions for Authors

MS has been significantly improved. The only thing that bothers me is that MS itself has poor quality drawings. However, you can see them in higher resolution. I would encourage authors to divide the drawings into at least 2 parts and enlarge them. MS would definitely gain in quality then. I think MS is really good and raises very interesting issues synchronously mature intersex in Japanese flounder.

Answer: Thanks for your suggestion. According to your suggestion, we divided figure. 1 into figure. 1 and figure. 2, and revised figure. 2 to figure. 3.